# Photoinitiated Polymerization of Hydrogels by Graphene Quantum Dots

**DOI:** 10.3390/nano11092169

**Published:** 2021-08-25

**Authors:** Yuna Kim, Jaekwang Song, Seong Chae Park, Minchul Ahn, Myung Jin Park, Sung Hyuk Song, Si-Youl Yoo, Seung Gweon Hong, Byung Hee Hong

**Affiliations:** 1Department of Chemistry Seoul National University, Seoul 08826, Korea; ykim2189@gmail.com (Y.K.); saver04@snu.ac.kr (J.S.); mincheol@snu.ac.kr (M.A.); hanson2525@gmail.com (M.J.P.); 2Graphene Research Center, Advanced Institute of Convergence Technology, Suwon 16229, Korea; 3Program in Nano Science and Technology, Graduate School of Convergence Science and Technology, Seoul National University, Seoul 08826, Korea; pscdaniel@snu.ac.kr; 4Department of Mechanical Engineering, Seoul National University, Seoul 08826, Korea; shksong01023@gmail.com; 5Interojo Inc., Pyeongtaek 17744, Korea; yoosy@interojo.com (S.-Y.Y.); steve.hong@interojo.com (S.G.H.)

**Keywords:** graphene quantum dots, hydrogel, organic-inorganic nanostructures, photoinitiator, photopolymerization

## Abstract

As a smart stimulus-responsive material, hydrogel has been investigated extensively in many research fields. However, its mechanical brittleness and low strength have mattered, and conventional photoinitiators used during the polymerization steps exhibit high toxicity, which limits the use of hydrogels in the field of biomedical applications. Here, we address the dual functions of graphene quantum dots (GQDs), one to trigger the synthesis of hydrogel as photoinitiators and the other to improve the mechanical strength of the as-synthesized hydrogel. GQDs embedded in the network effectively generated radicals when exposed to sunlight, leading to the initiation of polymerization, and also played a significant role in improving the mechanical strength of the crosslinked chains. Thus, we expect that the resulting hydrogel incorporated with GQDs would enable a wide range of applications that require biocompatibility as well as higher mechanical strength, including novel hydrogel contact lenses and bioscaffolds for tissue engineering.

## 1. Introduction

Hydrogels, neither classified as completely liquid nor as solid state, are hydrophilic polymer networks which can absorb water to their porous networks. They have been of great interest to researchers in various applications such as polymer contact lenses and tissue engineering. The three-dimensional porous network structure can be modulated by controlling the ratio of monomers and crosslinking agents during polymerization. Moreover, the stimulus-responsive volume changes sensitive to temperature, pH, solvent, and electric field enables them to be applied in various biomedical applications [1,2,3,4,5,6,7,8,9,10,11]. In addition, the excellent optical and mechanical properties of the hydrogels can be applied to multifunctional contact lenses [12,13,14].

However, there have been some limitations of hydrogels for practical use. Firstly, they show poor mechanical properties with low tensile/compression strength and toughness. [5,15,16] Second, the conventional photoinitiators needed for hydrogel photopolymerization have an innate toxicity, limiting its usage in biological applications [17,18]. Thus, several methods have been proposed to synthesize hydrogels with enhanced mechanical properties and biocompatibility by hybridizing different nanomaterials [19,20,21,22,23,24]. Several carbon-based nanocomposites, such as carbon nanotubes, graphene oxides (GOs), or functionalized graphene sheets, have been implemented to improve the mechanical properties. However, these carbon nanomaterials lack the ability to function as photoinitiators, and therefore, additional toxic substances need to be employed to generate the radicals to trigger the polymerization [16,22,25,26].

The photoinitiator should be a photosensitive molecule with an absorption wavelength range adequate for the initiation of polymerization reaction, and the absorbed photon should possess sufficient energy to generate free radicals. In this process, the functional groups on the initiator can expedite the polymerization reaction [27,28,29]. The graphene quantum dot (GQD), unlike the other graphene derivatives, is known to have distinctive optical properties showing size and edge-dependent fluorescence properties [24,30,31,32,33,34,35,36,37,38]. It shows wide absorption spectra ranging from UV to visible wavelengths. Furthermore, GQDs are expected to show better biocompatibility than other inorganic semiconductor nanoparticles such as TiO_2_ and ZnO when they are used solely or as a composite [32,39,40,41,42,43].

Thus, we employed the dual functionality of GQDs as photoinitiators for the polymerization of polyacrylamide hydrogels and mechanical reinforcers of as-synthesized hydrogel networks (Figure 1). They acted effectively even with sunlight as photoinitiators due to their broad absorption range, and achieved high Young’s moduli by up to 50 times. The swelling ratio was similar or slightly increased compared to the hydrogel fabricated by the conventional photoinitiator. Finally, we demonstrated a potential application for contact lenses with high transmittance (≥90%).

## 2. Materials and Methods

### 2.1. Synthesis of GQDs

Graphene quantum dots (GQDs) were synthesized by following the previously reported method [32,40,41,44]. In brief, 0.9 g of carbon fiber was added into an acidic mixture of concentrated H_2_SO_4_ and HNO_3_ in 3:1 (*v*/*v*) ratio. The mixed solution was sonicated for two hours and stirred for 24 h at 100 and 120 °C. The mixture was cooled and diluted with deionized (DI) water. The final product was then further dialyzed (molecular weight cut-off: 1 kDa) for 3 days, followed by lyophilization. Then, 0.01 g of as-fabricated GQD was mixed with 10 mL of DI water for its usage in hydrogel.

### 2.2. Characterization of GQDs

The GQD layers were characterized using various microscopic and spectroscopic techniques. Raman analysis was performed by inVia Raman Microscope (Renishaw, Gloucestershire, UK) An absorbance spectrum was obtained by UV−Vis-NIR spectrophotometer (S-3100, Scinco, Seoul, Korea). FT-IR spectra were acquired by FT-IR spectrophotometer (Nicolet 6700, Thermo Scientific, Waltham, MA, USA). The photoluminescence characterization was performed by the fluorescence spectrometer (FP-8300, Jasco Inc., Easton, MD, USA) with Xe lamp as the source of excitation.

### 2.3. Preparation of GQDGel and AGel

Acrylamide (AA) and *N*,*N*′-methylenebisacrylamide (MBAA) were dissolved in DI water with 40:1 and 400:1 of molar ratio. The solution was degassed in a vacuum chamber for 30 min. *N*,*N*,*N*′,*N*′-tetramethylethylenediamine (TEMED) was added into each solution, followed by addition of GQD solution (0.01 g/mL) or ammonium persulfate (APS) solution (0.01 g/mL). The solution was, finally, incubated under sunlight to be polymerized.

### 2.4. Dynamic Light Scattering (DLS) Analysis

The mixed solution was added to the cuvettes and exposed to the sunlight for polymerization. The measurement was carried out at room temperature with scattering angle of 275° by dynamic light scattering analyzer (Nano S, Malvern, Worcestershire, UK). The detailed calculation for DLS analysis is described in Appendix A.

### 2.5. Mechanical Characterization of Hydrogels

Stress−strain curves for Young’s modulus calculation were acquired by the universal testing system (INSTRON 5948, Instron, Norwood, MA, USA). A dog-bone shape of hydrogel specimen was fabricated for tensile test by ASTM D412. Hydrogel mixture solution was poured into the fabricated PDMS mold and left under the sunlight for polymerization. The tensile test for each specimen was performed with the 2 mm/min of extension. Surface adhesion force was measured by atomic force microscopy (XE-100, Park Systems, Suwon, Korea).

### 2.6. Electron Microscopy Analysis

Samples were lyophilized for scanning electron microscopy (SEM) observation. The dried AGel and GQDGel were cut into small pieces followed by Pt coating for SEM imaging. SEM observation was performed with 10 kV of operating voltage by field-emission SEM (JSM-6700F, JEOL Ltd., Tokyo, Japan). For tunneling electron microscopy (TEM) observation, a thin layer of solution was placed on TEM copper grid for direct polymerization and air-dried. TEM images were acquired by Cs-corrected TEM with Cold FEG (JEM-ARM200F, JEOL Ltd., Tokyo, Japan) with acceleration voltage of 80 kV.

## 3. Results and Discussion

First, several characterizations were performed to assure their photoresponsive properties and successful synthesis, such as Raman analysis, UV−Vis-NIR spectrophotometer, photoluminescence (PL) and Fourier-transform infrared (FT-IR) analysis (Figure 1). The Raman spectra of GQDs clearly showed two peaks, D (1350 cm^−1^) and G (1605 cm^−1^), indicating the successful synthesis of GQDs. In the absorbance spectrum, the peak around 250 nm was observed, which is related to π-π* transition of the conjugated C=C double bonds from graphitic characteristics of GQDs [45]. From the PL analysis, the emission peaks were observed at around 480 nm and 550 nm in all excitation range from 300 nm to 480 nm. In the FT-IR spectra, we could clearly observe several functional groups, such as carboxylic acid (1715 cm^−1^), aromatic double bond (1610 cm^−1^), ester (1236 cm^−1^), and aromatic ether (1132 cm^−1^); these functional groups are responsible for the high solubility of GQDs in water and for photoinitiation of polymerization purposes, as previously reported [32,40,41,44]. The wide range of photoresponsive properties of GQDs can be extensively used with a variety of light sources, even sunlight.

Light-responsive gelation with GQDs was confirmed by incubating the gelation solution either in a dark room or under sunlight. Incubation for 24 h in the dark room showed no indication of polymerization, but the sample after 20 min incubation under sunlight was transformed into the hydrogel (Figure 2a). In addition, we controlled the amount of GQD added in the gelation solution (Figure 2b,c). Ammonium persulfate (APS), a conventional photoinitiator for polymerization, was utilized to form hydrogels for comparison. All samples photoinitiated with GQDs and APS were transformed into hydrogels after exposure to sunlight as expected. On the other hand, no gelation was observed when GOs were employed as photoinitiators in the same conditions (Appendix A).

To confirm the sol−gel transition quantitively, dynamic light scattering (DLS) analysis for each sample was performed. DLS is one of the analysis tools for observing the dynamics or structural changes during gelation steps [46,47,48,49,50]. The detailed calculation method of DLS analysis for hydrogel is described in Appendix A. The time-averaged intensity correlation function (*g*^(2)^(τ)–1), gelation factor κ, and dynamic structure factor S(q,τ) were plotted for AGel and GQDGels at the specific exposure time under sunlight, and sol−gel transition was clearly observed (Figure 3, Appendix A). Additionally, sol−gel transition of GQDGel_0.1 was confirmed by scanning electron microscopy (SEM) observation by exposure time (Appendix A). As the incubation time increased, the macromolecular structures from sol to gel state were observed. The walls of the polymers became smooth as no more active sites remained or as the reaction went to termination.

To compare the microstructures of AGels with those of GQDGels, the cross-sectional view of AGel and GQDGel_0.2 samples were observed by SEM, which showed a highly porous structure with pore sizes ranging from a few to tens of micrometers (Figure 4). The pore size of the hydrogel was dependent on the ratio of monomer (AA) to crosslinking agent (MBAA); as the ratio of AA to MBAA increased, the pore size of the hydrogel decreased. It is because a higher concentration of crosslinkers produce a large degree of polymer chain branches, resulting in the generation of additional networks. In addition, the amount of GQD hardly affected the pore structure, but the walls of the polymer network became thicker with an increased concentration of GQDs.

To confirm the embedment of GQDs in GQDGels, we performed Raman analysis of GQDGels, but only the characteristic Raman peaks of polyacrylamide were observed (Appendix A). In order to directly isolate GQDs from GQDGels and run it through Raman analysis, we dissolved GQDGels with hydrogen peroxide solution, followed by dialysis and lyophilization (Appendix A). Transmission electron microscopy (TEM) images also supported that GQDs were randomly dispersed and embedded within the polymer structure (Appendix A).

To verify the role of GQDs as mechanical reinforcers in the hydrogel structure, the fabricated hydrogel samples were prepared in a dog-bone shape for the tensile test (Figure 5a, Appendix A). Young’s modulus of AGel slightly increased with the increased ratio of AA to MBAA component as previously reported, normally ranging from 10 to 30 kPa [51]. In contrast, the maximum Young’s modulus of GQDGel_1.0 with 40:1 (AA:MBAA) ratio was measured as approximately 550 kPa, which was about 50 times larger than that of AGel with the same ratio. Next, we measured the surface adhesion force by atomic force microscopy (AFM), in which the pull-off force during the retraction of the AFM tips can inform about the free chemical chains on the surface of hydrogel. (Figure 5b, Appendix A) As shown from the mapping of the force-displacement measurement, the average of maximum surface adhesion force increased as the concentration of GQD increased. This is due to the addition of GQDs strengthening the hydrogel and also enhancing the adhesion force that is normally restrained on the hydrogel surface [16,19,25].

Next, to define the fractional increase in the weight of the hydrogel due to water absorption, the swelling ratios of the fabricated hydrogel were measured (Figure 6). The swelling properties are usually affected by the network density, polymer-solvent interaction parameter, and by the pore size of the hydrogels [52,53,54]. The swelling ratio (Q) can be measured from the following Equation (1)
(1)$$Q=\;\frac{W_{s,t}\;-{\;W}_{d}}{W_{d}}\;\times 100\%,$$
where W_s,t_ is the weight of swollen hydrogel at time t and W_d_ is the weight of the dry hydrogel at t = 0. At the initial step, the swelling rate of GQDGel was about six times faster than that of AGel. We assume the hydrophilic functional groups on GQD contributed to increasing the swelling rate as well as the maximum swelling ratio of the hydrogel. The maximum swelling ratio of GQDGel was similar or slightly higher compared to AGel. However, when the excess amount of GQD was introduced during polymerization, the swelling ratio decreased. This is because the growing macromolecular chains are terminated at a faster rate as the concentration of initiator increases, resulting in the formation of low molecular-weight polymeric segments within the hydrogel network [52,54,55,56]. The excellent transparency as well as the controllable mechanical behavior and swelling property allowed our fabricated hydrogels to be potentially coapplied in the contact lenses or ocular drug delivery (Appendix A) [57,58,59].

## 4. Conclusions

In summary, we synthesized chemically flexible and mechanically tough hydrogels using GQDs as photoinitiating reagents and confirmed that the unique photochemical properties of GQDs trigger the efficient crosslinking process of acrylamide monomers, enhancing its mechanical strength and swelling behaviors. Unlike GQDs, graphene oxides (GOs) were found to be inactive for the photoinduced polymerization of hydrogels. The dynamic light scattering (DLS) and electron microscopy (EM) analyses of the gelation steps indicate that the hydrogels were synthesized as GQD-embedded polymer matrices. As a result, Young’s modulus and the surface adhesion the GQD-embedded hydrogels (GQDGel) were enhanced by 50 times and 2.5 times compared to the common AGel, respectively. The maximum swelling ratio of GQDGel was similar to that of AGel, but the excessive amount of GQDs that induced fast termination of polymerization reactions resulted in hardening and less swelling behaviors. We also demonstrated the potential application of the GQDGels to advanced contact lenses. In addition, the biocompatible GQD-based hydrogels with controllable mechanical properties and pore sizes as well as fluorescent properties would be useful for tissue engineering and drug delivery systems in the future.

## Data Availability

The data presented in this study are available on request from the corresponding author.

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
