# Peer review of "Photoinitiated Polymerization of Hydrogels by Graphene Quantum Dots"

_nanomaterials, 2021, doi:10.3390/nano11092169_

Round 1

Author Response

Please find the attached file for the response to the reviewer 1's comments.

Reviewer 2 Report

In this manuscript, the author reports, ‘Photo-Initiated Polymerization of Hydrogels by Graphene Quantum Dots’. The current study is on a topic of relevance and general interest to readers in this area. The authors should address the following questions before getting a possible publication.

 Recommendation: Major revisions needed as noted.

  1. In the ‘Synthesis of GQDs’ section, the author should mention the molecular cutoff of the dialysis membrane used.
  2. In Page 3, line no 120-123, the author should mention in the text the peak values of the different functional groups in the FTIR spectra.
  3. The author should write the purpose for each test in one/two sentences (in brief) before explaining the results of the characterization techniques. Therefore, the logic and organization of this part will be enhanced
  4. What does the error bars stand for presented in the Figure 5 and Figure 6b? It should be mentioned in Figure captions.
  5. The novelty of the present work should be discussed in the Introduction section.
  6. The formatting and grammatical errors in the article need to be checked carefully.
  7. The authors have cited relevant references in the Introduction section; however there are few that need to be included: Analytica Chimica Acta 974 (2017): 93-99; ACS Applied Materials & Interfaces, 12(46), 51940-51951; Langmuir, 33(43), 12344-12350; Research on Chemical Intermediates 45.7 (2019): 3823-3853;

Author Response

Please find the attached file for the response to the reviewer 2's comments.

Round 2

Reviewer 1 Report

  1. The authors identified in the FTIR spectrum carboxylic acid as due to functionalized GQD having carboxylic acid group attached. In the manuscript in Scheme 1, it shows normal GQD. The authors should state that GQD-functionalized carboxylic acid. The authors should add reference they cited as an explanation in the referee’s comments. J. Membrane Sci. 2018, 556, 293-302, https://doi.org/10.1016/j.memsci.2018.04.009
  2.  The authors should include discussion about GQD identification by Raman spectroscopy; the difficulties associated in characterizing the gel

Author Response

Please find the attached file for the 2nd response to the reviewer's comments.

Reviewer 2 Report

The authors have addressed all the questions raised previously. The manuscript can be accepted now.

Author Response

Thank you very much for the comment that recommends the publication of the manuscript.